# Predicting Adverse Recanalization Therapy Outcomes in Acute Ischemic Stroke Patients Using Characteristic Gut Microbiota

**DOI:** 10.3390/microorganisms11082016

**Published:** 2023-08-05

**Authors:** Ping-Song Chou, Wei-Chun Hung, I-Hsiao Yang, Chia-Ming Kuo, Meng-Ni Wu, Tzu-Chao Lin, Yi-On Fong, Chi-Hung Juan, Chiou-Lian Lai

**Affiliations:** 1Department of Neurology, Kaohsiung Medical University Hospital, Kaohsiung Medical University, Kaohsiung 807378, Taiwan; pschou1013@gmail.com (P.-S.C.); berkeley0701@gmail.com (M.-N.W.); prochristy@gmail.com (T.-C.L.); amyfong0813@gmail.com (Y.-O.F.); 2Department of Neurology, Faculty of Medicine, College of Medicine, Kaohsiung Medical University, Kaohsiung 807377, Taiwan; 3Graduate Institute of Medicine, College of Medicine, Kaohsiung Medical University, Kaohsiung 807377, Taiwan; 4Neuroscience Research Center, Kaohsiung Medical University, Kaohsiung 807377, Taiwan; 5Department of Microbiology and Immunology, Kaohsiung Medical University, Kaohsiung 807377, Taiwan; wchung@kmu.edu.tw; 6Department of Medical Imaging, Kaohsiung Medical University Hospital, Kaohsiung Medical University, Kaohsiung 807378, Taiwan; toni_yihsiao@yahoo.com.tw; 7Department of Nursing, Kaohsiung Medical University Hospital, Kaohsiung Medical University, Kaohsiung 807378, Taiwan; 960162@mail.kmuh.org.tw; 8Institute of Cognitive Neuroscience, National Central University, Taoyuan City 320, Taiwan; chihungjuan@gmail.com; 9Cognitive Intelligence and Precision Healthcare Research Center, National Central University, Taoyuan City 320, Taiwan

**Keywords:** acute ischemic stroke, endovascular thrombectomy, gut microbiota, intravenous thrombolysis

## Abstract

Recanalization therapy is the most effective treatment for eligible patients with acute ischemic stroke (AIS). Gut microbiota are involved in the pathological mechanisms and outcomes of AIS. However, the association of gut microbiota features with adverse recanalization therapy outcomes remains unclear. Herein, we investigated gut microbiota features associated with neurological deficits in patients with AIS after recanalization therapy and whether they predict the patients’ functional outcomes. We collected fecal samples from 51 patients with AIS who received recanalization therapy and performed 16S rRNA gene sequencing (V3–V4). We compared the gut microbiota diversity and community composition between mild to moderate and severe disability groups. Next, the characteristic gut microbiota was compared between groups, and we noted that the characteristic gut microbiota in patients with mild to moderate disability included *Bilophila*, *Butyricimonas*, Oscillospiraceae_*UCG-003*, and *Megamonas*. Moreover, the relative abundance of *Bacteroides fragilis*, *Fusobacterium* sp., and *Parabacteroides gordonii* was high in patients with severe disability. The characteristic gut microbiota was correlated with neurological deficits, and areas under the receiver operating characteristic curves confirmed that the characteristic microbiota predicted adverse recanalization therapy outcomes. In conclusion, gut microbiota characteristics are correlated with recanalization therapy outcomes in patients with AIS. Gut microbiota may thus be a promising biomarker associated with early neurological deficits and predict recanalization therapy outcomes.

## 1. Introduction

Stroke is a brain injury caused by a disruption in the blood supply to a specific region of the brain, resulting in permanent neurological deficits or even death. Stroke is not only a major global health issue but also the fourth leading cause of death and a significant contributor to disability in Taiwan. It can be classified into ischemic and hemorrhagic stroke. Approximately 87% of stroke cases are of the ischemic type [1]. Recanalization therapy, which includes intravenous thrombolysis (IVT) and endovascular thrombectomy (EVT), is currently recommended by treatment guidelines for patients with acute ischemic stroke (AIS) who meet the selection criteria [2]. However, most patients with AIS remain dependent in their activities of daily living. Therefore, investigating prognostic factors that potentially predict recanalization therapy outcomes in patients with AIS is of clinical significance. Several predictors for functional outcomes after recanalization therapy for AIS have been reported; they include age, atrial fibrillation, initial stroke severity, onset to treatment time, cerebral collateral flow, and the number of EVT passes [3].

In addition to clinical factors, the role of inflammatory reactions in the prognosis of AIS is a crucial area of investigation. Neuroinflammatory cascades are activated early, and their progression is rapid after AIS [4]. The mechanisms of neuroinflammation-driven injury during the acute phase of AIS involve several factors, including excessive oxidative stress, increased production of matrix metalloproteinase, activation of microglia and astrocytes, release of proinflammatory cytokines, and migration of immune cells into the ischemic area [5,6]. This cascade can lead to disruption of the blood–brain barrier (BBB), neuronal injury, cerebral edema, hemorrhagic transformation, and a worsened neurological outcome [6,7]. Furthermore, following recanalization therapy for AIS, oxidative stress, mitochondrial dysfunction, calcium overload, and excitotoxicity are dramatically increased due to the reperfusion injury. This, in turn, potentiates the extent of neuroinflammation and brain tissue damage caused by the initial ischemic insult [8,9]. Consequently, counteracting neuroinflammation should be considered as an adjunct therapeutic strategy in AIS patients, which could extend the range of the clinical benefits of recanalization therapy [10].

The gut microbiota—referring to the collection of bacteria, archaea, and eukaryotes colonizing the gastrointestinal tract—is a key regulator of the poststroke neuroinflammatory response [11,12]. Patients with AIS demonstrate dysbiosis; that is, a substantial change in gut microbiota diversity and abundance and an increase in the number of pathogenic bacteria [13]. The microbiota–gut–brain axis involves bidirectional communication between the gut and the brain. Through top–down communication, AIS affects gut function and gut microbial composition through the autonomic nervous system, enteric nervous system, and hypothalamic–pituitary–adrenal axis. AIS disrupts intestinal motility, leads to increased intestinal permeability, reduces mucus secretion from goblet cells, and contributes to dysbiosis. In bottom–up communication, dysbiotic gut bacteria produce endotoxins (e.g., lipopolysaccharide, LPS) and metabolites (e.g., short-chain fatty acids (SCFA) and trimethylamine-N-oxide), which activate resident immune cells, trigger inflammatory responses, and secrete proinflammatory cytokines. Immune cells migrate to the ischemic area of the brain, and inflammatory mediators reach the brain through systemic circulation, leading to the exacerbation of neuroinflammation and ischemic injury after AIS [12,14]. Dysbiosis is correlated with poor functional outcomes after AIS [15,16,17].

Modulation of the gut microbiota may improve functional outcomes after AIS [18]. However, clinical studies delineating the potential association between gut microbiota composition and recanalization therapy outcomes in patients with AIS are lacking. This is mainly because recanalization therapy is time-limited, and only a few patients with AIS are eligible for it. Moreover, post-AIS changes in physical activity and diet may influence gut microbiota composition.

Gut microbiota composition may differ among patients with AIS with different functional outcomes after recanalization therapy. Therefore, in this prospective, observational study, we compared the gut microbiota profiles of patients with AIS who received recanalization therapy through 16S ribosomal RNA (16S rRNA) gene sequencing. Our results may provide insights into the factors predicting the post-recanalization therapy prognosis in patients with AIS.

## 2. Materials and Methods

### 2.1. Patients

This was a single-center, prospective cohort study, and it was conducted at a hospital in the southern part of Taiwan. As indicated in Figure 1, patients who were admitted to the hospital within 6 h of symptom onset and given a diagnosis of AIS were enrolled. We included only patients who received recanalization therapy (i.e., IVT, EVT, or both) as well as brain magnetic resonance imaging to detect acute ischemic lesions. We excluded patients who were younger than 20 years, had intracerebral hemorrhage on initial radiological examination, had contraindications to IVT or EVT, had received probiotics or antibiotics within 1 week prior to admission, or had received antibiotic therapy prior to fecal sample collection after admission. Each patient was comprehensively evaluated for demographics, medical history, physical and neurological examinations, and blood biochemistry analysis.

### 2.2. Stroke Severity, Functional Outcomes, and Reperfusion Assessment

Stroke severity was evaluated on the basis of each patient’s National Institutes of Health Stroke Scale (NIHSS) score, whereas functional outcomes were assessed using each patient’s modified Rankin Scale (mRS) score. A neurologist blinded to the results of the gut microbiota analyses assessed both the NIHSS and mRS scores. The NIHSS score (range of 0–42) indicates the degree of a patient’s neurological impairment; the higher the NIHSS score, the more severe the patient’s neurological deficit. The NIHSS score was obtained before treatment and at discharge. The mRS score was assessed at baseline and at discharge; an mRS score of 0–3 at discharge was considered to indicate mild to moderate disability. Post-EVT reperfusion status was assessed using the modified treatment in cerebral ischemia (mTICI) score. The mTICI score was determined by the EVT operator based on each patient’s final angiogram.

### 2.3. Fecal Sample Collection, Bacterial DNA Extraction, and 16S rRNA Gene Sequencing

Fecal samples were collected from each participant before their first meal during hospitalization. The samples were frozen immediately after collection and delivered to the laboratory in a cooler bag within 24 h. The fecal samples were stored at −80 °C for up to 3 days prior to processing.

Bacterial deoxyribonucleic acid (DNA) was extracted from the fecal samples using a stool DNA extraction kit (Topgen Biotechnology, Kaohsiung, Taiwan). After its quality and concentration were assessed on a Colibri Microvolume spectrophotometer (Titertek Berthold, Pforzheim, Germany), the extracted DNA was immediately frozen at −20 °C.

We outsourced our DNA samples to Welgene Biotech (Taipei, Taiwan) for 16S rRNA gene sequencing. Each bacterial DNA sample was subjected to 16S rDNA amplicon sequencing using Illumina Sequencing-by-Synthesis technology on an Illumina MiSeq sequencer to produce 2 × 300 bp paired-end reads. The primers for the 16S rRNA gene (V3–V4 region) were as follows: forward, TCGTCGGCAGCGTCAGATGTGTATAAGAGACAGCCTACGGGNGGCWGCAG; and reverse, GTCTCGTGGGCTCGGAGATGT GTATAAGAGACAGGACTACHVGGGTATCTAATCC.

### 2.4. Statistical and Bioinformatic Analyses of Microbiota

The patients’ demographic, medical history, and blood biochemistry data were classified as either categorical or continuous variables. In order to compare the data between patients with mild to moderate disability (mRS score = 0–3) and severe disability (mRS score = 4–6), we utilized the two-tailed independent *t*-test to analyze continuous variables, while categorical variables were assessed using the chi-square test. The NIHSS scores, type of recanalization therapy, onset to treatment time, presence of good angiographic reperfusion (mTICI score ≥ 2b), and occluded artery were also compared between the two groups.

The raw sequencing data were imported into QIIME2 [19] and processed using the DADA2 plugin [20] to merge and denoise paired-end reads into amplicon sequence variants (ASVs). The median number (interquartile range (IQR)) of reads filtered through each quality control step was 105,487 (95,432, 117,353). To avoid false conclusions due to an uneven sampling depth in the microbiome diversity assessment, we standardized the sampling depth of each sample by rarefying it to 66,649 reads, which corresponded to the lowest number of reads detected in all samples and the point at which the rarefaction curves of both groups leveled off.

We compared the alpha diversity indexes using pairwise Kruskal–Wallis tests. To assess beta diversity, we performed a pairwise analysis of similarities (ANOSIM) and permutational multivariate analyses of variance (PERMANOVA) with 999 permutations as well as a principal coordinate analysis (PCoA) based on various distance matrixes. All *p* values were adjusted using the Benjamini–Hochberg procedure (to obtain *q* values).

The ASV taxonomy was classified using a SciKit Learn-based approach and by searching in the SILVA reference database (version 138; trimmed to the V3–V4 region; L7 taxonomy) [21]. We analyzed the relative abundance of taxa using linear discriminant analysis (LDA) effect size (LEfSe) [22]. Next, we identified differential taxa features between the groups, which were identified on the basis of a log LDA score for discriminative features of >2 and *p* < 0.05 in the factorial Kruskal–Wallis test.

The Spearman correlation was used to analyze the associations between gut microbiota and stroke severity, as determined by NIHSS. To explore the value of gut microbiota features in the prediction of post-recanalization therapy outcomes, we plotted receiver operating characteristic (ROC) curves based on the relative abundance of the bacteria in the two patient groups and calculated the areas under ROC curves (AUCs).

### 2.5. Ethics Approval

This study was conducted in accordance with the guidelines of the Declaration of Helsinki and approved by the Institutional Review Board of Kaohsiung Medical University Hospital [KMUHIRB-E(I)-20200424]. Informed consent was obtained from all patients involved in this study or their legal representatives.

## 3. Results

### 3.1. Patient Characteristics

A total of 51 fecal samples were collected from patients undergoing recanalization therapy for AIS. The mean age of the patients was 70.6 (±13.2) years, and 29 (56.9%) were men. At admission, the median (IQR) NIHSS score was 15 (10, 21). Of the patients, 30 (58.8%) received IVT, 13 (25.5%) received EVT, and 8 (15.7%) received both. The patients were discharged a median (IQR) of 19 (12, 36) days after admission; their median (IQR) NIHSS score was 5 (2, 15). Overall, 40 (78.4%) patients showed neurological improvement, and 23 (45.1%) were classified as having mild to moderate disability. Patients with mild to moderate disability had significantly lower median NIHSS scores at admission and discharge than those with severe disability.

Compared with the severe disability group, the mild to moderate disability group had a significantly lower atrial fibrillation or flutter prevalence and a significantly higher rate of only IVT use. However, the between-group differences in the median time from stroke onset to IVT, groin puncture, or reperfusion and in the rates of successful reperfusion (mTICI score ≥ 2b) were nonsignificant. Most (84.3%) of the occluded vessels were in the anterior circulation; the proportions of the patients in each group with such vessels were similar. Table 1 presents an overview of the patient characteristics, vascular risk factor, and administration details of the recanalization therapy.

### 3.2. Characterization of Gut Microbiota Based on 16S rRNA Gene Sequencing

As presented in Figure 2, intragroup diversity tended to be lower in the severe disability group than in the mild to moderate disability group, as indicated by the Chao1 index (586.69 ± 101.67 vs. 534.00 ± 114.07, *p* = 0.091) and Shannon’s index (5.9 ± 0.6 vs. 5.7 ± 0.7, *p* = 0.391) values; however, these differences were nonsignificant. Regarding the intergroup diversity, PCoA based on Bray–Curtis dissimilarity (ANOSIM: R = −0.02, *p* = 0.74; PERMANOVA: pseudo-F = 1.00, *p* = 0.46) and weighted unnormalized UniFrac (ANOSIM: R = −0.03, *p* = 0.90; PERMANOVA: pseudo-F = 0.59, *p* = 0.87) demonstrated an absence of significant microbial clustering differences between the mild to moderate and severe disability groups. These results reveal no significant differences in the intragroup and intergroup diversity between our mild to moderate and severe disability groups.

### 3.3. Relative Abundance of Discriminative Taxa between Mild to Moderate Disability and Severe Disability Groups

Between-group differences in the relative abundance of gut microbiota were estimated using LEfSe based on a log LDA score of >2. The differences in the abundance of the phyla Firmicutes (49.69% in the mild to moderate disability group vs. 50.16% in the severe disability group), Bacteroidetes (29.72% in the mild to moderate disability group vs. 29.41% in the severe disability group), and the Firmicutes/Bacteroidetes ratio (2.68 ± 3.43 in the mild to moderate disability group vs. 2.25 ± 1.68 in the severe disability group) were nonsignificant.

Figure 3 presents the abundant taxa among the patient groups. The Oscillospiraceae_*UCG-003* and *Megamonas*, as well as its family Selenomonadaceae in the phyla Firmicutes, *Butyricimonas* and its family Marinifilaceae, *Bacteroides fluxus* and *Alistipes shahii* in the phyla Bacteroidetes, *Bifidobacterium* sp., and *Bilophila* were significantly enriched in the mild to moderate disability group. In contrast, *Bacteroides fragilis* and *Parabacteroides gordonii* in the phyla Bacteroidetes and *Fusobacterium* sp. were significantly enriched in the severe disability group. Random forest models were used for taxonomy prediction, and four genera could be used to discriminate the mild to moderate disability group from the severe disability group: *Bilophila*, *Butyricimonas*, Oscillospiraceae_*UCG-003*, and *Megamonas*.

### 3.4. Analysis of Association between Gut Microbiota, NIHSS Scores, and Functional Outcomes

We selected *Bilophila*, *Butyricimonas*, Oscillospiraceae_*UCG-003*, *Megamonas*, *Bacteroides fragilis*, *Fusobacterium* sp., and *Parabacteroides gordonii* for further analysis of the associations among gut microbiota, stroke severity, and functional outcomes based on LDA values and random forest models. As indicated in our Spearman correlation heatmap (Figure 4), the discharge NIHSS was correlated negatively with *Bilophila* (*p* = 0.004) and *Megamonas* (*p* = 0.011) but positively correlated with *Bacteroides fragilis* (*p* = 0.037). Moreover, *Bilophila* was correlated with neurological improvement, as indicated by NIHSS score changes between discharge and admission (*p* = 0.032).

We subsequently assessed the potential of using gut microbiota as a biomarker for predicting recanalization therapy outcomes. As presented in Figure 5, *Bilophila* and *Butyricimonas* have good predictive power for mild to moderate disability (AUCs = 0.713 and 0.741, respectively), and *Bacteroides fragilis* and *Parabacteroides gordonii* have a good predictive power for severe disability (AUCs = 0.712 and 0.679, respectively). Therefore, the identified bacteria could be potential biomarkers for recanalization therapy outcomes in patients with AIS.

## 4. Discussion

To the best of our knowledge, this is the first study to delineate gut microbiota features and their associations with recanalization therapy outcomes in patients with AIS. Our results demonstrate that after recanalization therapy, gut microbiota composition differs between patients with mild to moderate and severe disability after AIS. We discovered that *Bilophila*, *Butyricimonas*, Oscillospiraceae_*UCG-003*, and *Megamonas* are enriched in patients with mild to moderate disability, whereas *Bacteroides fragilis*, *Fusobacterium* sp., and *Parabacteroides gordonii* are enriched in patients with severe disability. The richness of specific gut microbiota was noted to be correlated with neurological deficits post-recanalization therapy. Thus, *Bilophila* and *Butyricimonas* may predict mild to moderate disability, whereas *Bacteroides fragilis* and *Parabacteroides gordonii* may predict severe disability. Taken together, these findings indicate that gut microbiota are ideal, noninvasive fecal biomarkers for the early prediction of neurological deficits and functional outcomes in patients with AIS after recanalization therapy.

*Bilophila*, which is enriched in patients with stroke [23] and acute coronary syndromes [24], is associated with the consumption of animal protein and a lack of plant-based protein sources [25]. However, the pathological mechanism underlying the association between *Bilophila* and AIS outcomes has not been established. The genera *Butyricimonas* and *Megamonas* and the family Oscillospiraceae can improve stroke outcomes through several mechanisms. *Butyricimonas* and Oscillospiraceae both produce butyrate [26,27,28], which has been noted to reduce neuronal apoptosis occurrence and cerebral infarction volume and to improve neurological function in animal stroke models [29,30]. In addition, *Megamonas* ferments glucose into short-chain fatty acids, mostly acetate and propionate [31], both of which are beneficial for stroke recovery. Reduced acetate and propionate levels were associated with an increased risk of poor functional outcomes in patients after stroke [16]. The results of an animal experiment demonstrated that supplementation with a mix of acetate, butyrate, and propionate improves poststroke recovery and cortical reorganization [32].

*Butyricimonas* can activate glucagon-like peptide-1 receptor and peroxisome proliferator-activated receptor α, which can alleviate diabetes and metabolic disorders induced by a high-fat diet [33]. Oscillospiraceae was reported to be correlated with adiponectin in neurodegeneration disease [34], and an abundance of Oscillospiraceae is associated with decreased insulin resistance [35]. The abundance of *Megamonas* is higher in individuals with normal glucose tolerance than in those with type 2 diabetes mellitus [36]. In general, the presence of *Butyricimonas*, Oscillospiraceae, and *Megamonas* can help stabilize glucose metabolism and increase short-chain fatty acid levels, potentially leading to improved stroke recovery and decreased disability.

On the other hand, *Bacteroides fragilis* can biosynthesize and secrete pathogenic and proinflammatory neurotoxins, namely LPS and *Bacteroides fragilis* toxins [37]. *Bacteroides fragilis* negatively affects the biophysiological barrier structure and function, and thus disrupts the normal blood–brain barrier and elicits inflammatory neuronal dysfunction [38]. Additionally, *Bacteroides fragilis* deteriorates glucose and lipid metabolism, activates an inflammatory response, and promotes atherosclerosis progression in animal models [39].

*Fusobacterium* generates a proinflammatory microenvironment in the gut [40], induces immune cell death [41], alters vascular endothelial integrity, and passes through the blood–brain barrier [42], eventually impairing stroke outcomes through the microbiota–gut–brain axis. Numerous studies have indicated that an increase in the number of *Fusobacterium* is associated with hypertension [43,44] and is positively correlated with homocysteine levels [45]—both of which are well-known risk factors for stroke. Therefore, increased *Fusobacterium* is considered to be strongly associated with unfavorable stroke outcomes [46].

*Parabacteroides*, a large artery atherosclerotic stroke biomarker [47], is more abundant in patients with ischemic stroke than in healthy individuals [15,48,49]. In addition, *Parabacteroides* is associated with vascular risk factors and stroke severity—as reflected by its positive correlation with infract volume and its negative correlation with poststroke daily function [47].

Taken together, our results indicate that in patients with AIS, a significant abundance of *Bilophila*, *Butyricimonas*, Oscillospiraceae, and *Megamonas*, which produce short-chain fatty acids and contribute to glucose homeostasis, possibly contributes to the beneficial effects of recanalization therapy. By contrast, *Bacteroides fragilis*, *Fusobacterium*, and *Parabacteroides* are associated with vascular risk factors, gut integrity disruption, blood–brain barrier impairment, and neuroinflammation induction, thereby increasing the likelihood of severe post-AIS disability.

There is a higher prevalence of atrial fibrillation/flutter observed in the severe disability group compared to the mild to moderate disability group among the demographics and medical history analyzed. Studies have shown that dysbiosis is linked to atrial fibrillation, possibly due to dietary habits, bacterial LPS, and microbial metabolites. These mediators are suggested to increase inflammation and contribute to atrial arrhythmogenesis, thereby affecting susceptibility to atrial fibrillation [50]. Furthermore, AIS patients with atrial fibrillation have reported unfavorable functional outcomes following IVT and EVT [51,52]. Hence, it is possible that in the severe disability group, gut microbiota may contribute to adverse functional recovery post-recanalization therapy via atrial fibrillation.

The novelty of the current study lies in its recruitment of patients with AIS who received recanalization therapy; thus far, this patient group has rarely been studied. This is because this population is small among AIS patients, as recanalization therapy is an urgent treatment strategy. We thus obtained a newer understanding of the use of gut microbiota as a prognostic biomarker of recanalization therapy outcomes in patients with AIS than other studies have.

Furthermore, targeting dysbiosis of the gut microbiota can potentially serve as a therapeutic intervention to alleviate poststroke neuroinflammation and to enhance stroke outcomes following recanalization therapy. An animal model revealed that the transplantation of healthy and SCFAs-producing microbiota notably improved stroke outcomes [11,29], while the modulation of the microbiota has been linked to a decrease in LPS and stroke-related neuroinflammation [53]. Additionally, a cerebral ischemia reperfusion model, similarly to recanalization therapy, demonstrated that microbiota from young mice may inhibit interleukin-17 production and lower reperfusion injury in aged mice [54].

In our view, although the majority of evidence comes from animal studies, microbiota-targeted therapy presents a promising potential for the treatment of AIS, particularly in patients undergoing recanalization therapy. It is conceivable that a therapy to modify the microbiota composition, such as dietary regulation, the administration of probiotics or prebiotics, and fecal microbiota transplantation, could be combined with recanalization therapy to mitigate the extent of reperfusion injury and neuroinflammation in the acute phase of AIS [55]. Thus, future clinical investigations are needed to explore the feasibility of targeting the gut microbiota as an innovative therapeutic approach that can improve functional outcomes in individuals with AIS.

The current study, however, has several limitations. First, no information regarding the patients’ dietary habits and lifestyles was collected. Patients who had used probiotics or antibiotics in the week before AIS diagnosis were excluded, and the fecal samples were collected before their first meal after receiving an AIS diagnosis. Through this design, we minimized the effects of diet and antibiotics on the gut microbiota after AIS. Second, we collected fecal samples at a single timepoint; this limited our ability to assess dynamic changes in the association of gut microbiota with functional outcomes after recanalization therapy. Third, because of our limited sample size, we could not determine the association of characteristic microbiota with their metabolites, such as trimethylamine-N-oxide and short-chain fatty acids. Thus, we could not investigate the causal association between gut microbiota and functional outcomes after recanalization therapy. Finally, our follow-up period was short. We determined functional outcomes on the basis of the mRS scores at discharge, mostly within 30 days. Future larger-scale and longer-term studies addressing these limitations and investigating the influence of gut microbiota on recanalization therapy outcomes by assessing the effects of metabolic products and pathways are warranted.

## 5. Conclusions

Our results confirmed the associations between gut microbiota characteristics and recanalization therapy outcomes in patients with AIS. Gut microbiota could be a pertinent biomarker for predicting adverse recanalization therapy outcomes in patients with AIS. Given the growing preclinical evidence suggesting that modulation of the gut microbiota is a promising therapeutic target for AIS, the translation of these results into clinical practice may represent a major breakthrough in the treatment of AIS and merits further research.

## Figures and Tables

**Figure 1 microorganisms-11-02016-f001:**
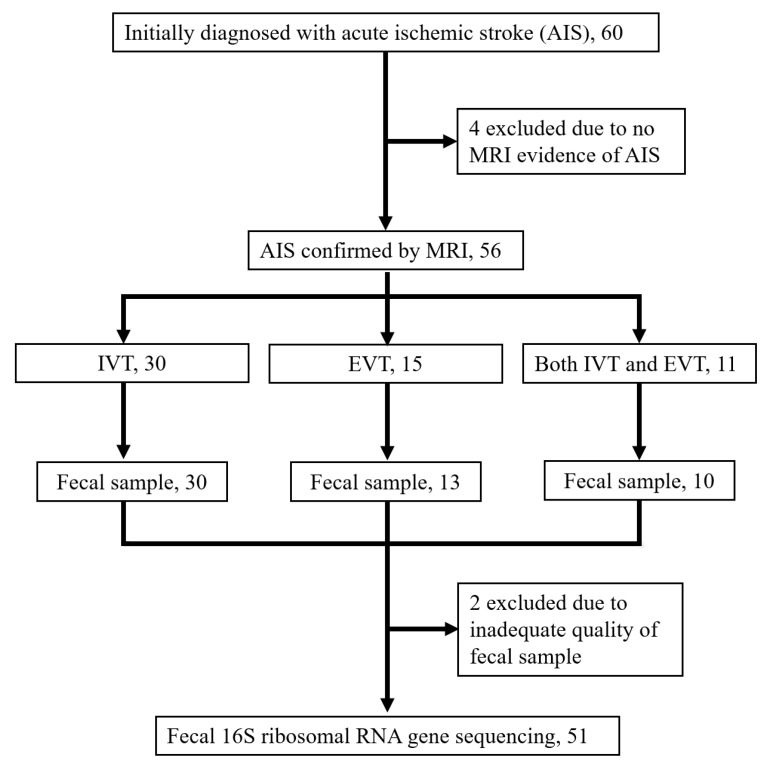
Participant enrollment and sample collection flow.

**Figure 2 microorganisms-11-02016-f002:**
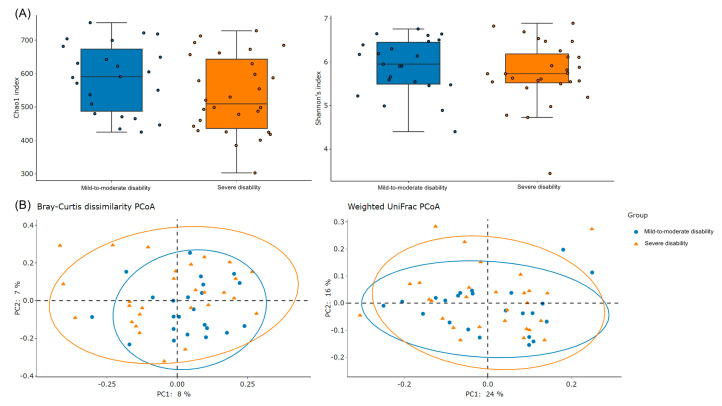
(**A**) Analysis and comparison of alpha diversity (i.e., Chao1 and Shannon’s indexes) in gut microbiota between mild to moderate and severe disability groups after recanalization therapy for acute ischemic stroke. (**B**) Principal coordinate analysis plot of gut microbiota based on Bray–Curtis and weighted UniFrac matrixes.

**Figure 3 microorganisms-11-02016-f003:**
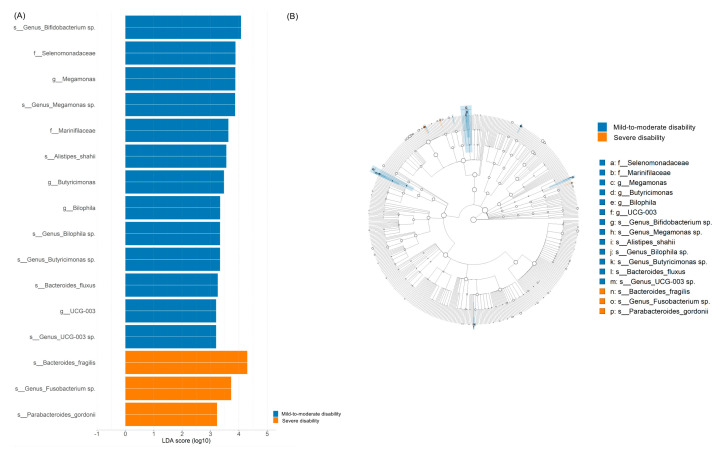
(**A**) Histogram of linear discriminant analysis (LDA) scores revealing the most differentially abundant taxa between mild to moderate and severe disability groups. Bacterial taxa with LDA score > 2 are presented. (**B**) Circular cladogram of LDA effect size analysis revealing bacteria with significant between-group differences.

**Figure 4 microorganisms-11-02016-f004:**
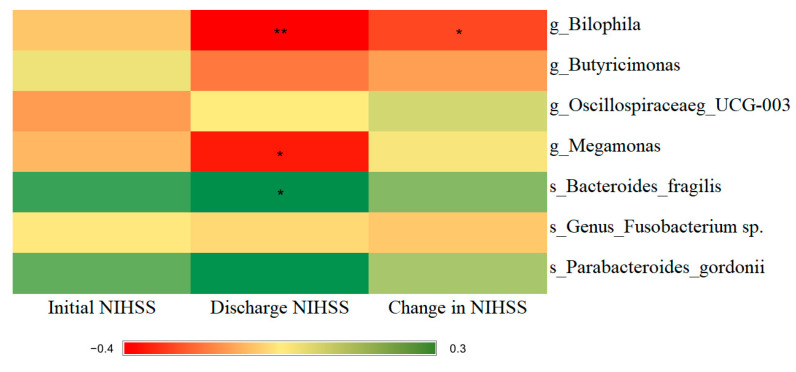
Heatmap of the correlation of the differential bacteria with NIHSS scores at admission and discharge and NIHSS score changes during hospitalization. Green grids represent positive Spearman’s rank correlation coefficients, whereas red grids represent negative Spearman’s rank correlation coefficients. The deeper green or red indicate higher correlation values. * and ** denote *p* < 0.05 and *p* < 0.01, respectively.

**Figure 5 microorganisms-11-02016-f005:**
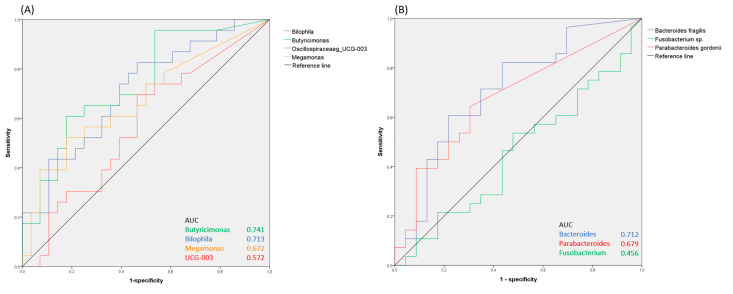
ROC curves indicating the predictive value of specific bacteria for functional outcomes in (**A**) mild to moderate and (**B**) severe disability groups.

**Table 1 microorganisms-11-02016-t001:** Participant characteristics.

Characteristic	Total (*n* = 51)	Mild to Moderate Disability (*n* = 23)	Severe Disability (*n* = 28)	*p*-Value
Age, years, mean (±SD)	70.6 ± 13.2	69.0 ± 12.7	72.0 ± 13.7	0.418
Sex, man, *n* (%)	29 (56.9%)	13 (56.5%)	16 (57.1%)	0.964
BMI, mean (±SD)	25.8 ± 3.5	26.3 ± 3.6	25.4 ± 3.3	0.371
Prior vascular risk factors				
Hypertension, *n* (%)	41 (80.4%)	16 (69.6%)	25 (89.3%)	0.078
Diabetes mellitus, *n* (%)	22 (43.1%)	8 (34.8%)	14 (50.0%)	0.275
Hyperlipidemia, *n* (%)	39 (76.5%)	19 (82.6%)	20 (71.4%)	0.349
Prior stroke, *n* (%)	10 (19.6%)	3 (13.0%)	7 (25.0%)	0.285
Atrial fibrillation/flutter, *n* (%)	24 (47.1%)	7 (30.4%)	17 (60.7%)	0.031 *
Smoking, *n* (%)	10 (19.6%)	2 (8.7%)	8 (28.6%)	0.075
Median initial NIHSS (IQR)	15 (10, 21)	10 (6, 15)	17.5 (13, 21)	0.012 *
Median discharge NIHSS (IQR)	5 (2, 15)	2 (1, 4)	13 (8.25, 24.5)	<0.001 **
Recanalization therapy, *n* (%)				
IVT only	30 (58.8%)	18 (78.3%)	12 (42.9%)	0.027 *
EVT only	13 (25.5%)	4 (17.4%)	9 (32.1%)	
Both IVT and EVT	8 (15.7%)	1 (4.3%)	7 (25.0%)	
Median onset to IVT time, minutes, IQR (*n* = 38)	115 (95, 164)	110 (95, 199)	119 (95, 150)	0.745
Median onset to puncture time, minutes, IQR (*n* = 21)	299 (252.5, 357.5)	255 (185, 384)	302 (273.25, 353.75)	1.000
Median onset to reperfusion time, minutes, IQR (*n* = 21)	370 (282.5, 427.5)	280 (202, 420.5)	379.5 (304, 438.75)	0.311
mTICI ≥ 2b, n (%) (*n* = 21)	19 (90.5%)	5 (100.0%)	14 (87.5%)	0.406
Occluded vessel, *n* (%)				0.762
Anterior circulation	43 (84.3%)	19 (82.6%)	24 (85.7%)	
Posterior circulation	8 (15.7%)	4 (17.4%)	4 (14.3%)	

BMI, body mass index; EVT, endovascular thrombectomy; IQR, interquartile range; IVT, intravenous thrombolysis; mTICI, modified treatment in cerebral ischemia; NIHSS, National Institute of Health Stroke Scale; SD, standard deviation. * *p* < 0.05. ** *p* < 0.01.

## Data Availability

The data presented in this study are available upon request from the corresponding author. The data are not publicly available due to privacy.

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
