# Peer review of "Predicting Adverse Recanalization Therapy Outcomes in Acute Ischemic Stroke Patients Using Characteristic Gut Microbiota"

_microorganisms, 2023, doi:10.3390/microorganisms11082016_

Round 1

Reviewer 1 Report

Dear editor thank you to allow me to read this original paper. 

The role of the gut microbiota in neurodegenerative diseases like parkinson disease or alzheimer disease it's known. There are no studies demonstrating this role in the pathogenesis of stroke. 

Instead contless studies showed the crucial role of neuroinflammation in stroke. 

In this way, this paper is very original and interesting. 

I suggest to expand in the introduction part that concern the neuroinflammation in stroke pathogenesis and prognosis. Add the correct refernces. 

The figure num. 4 is not really simple to decode. Please change it with another more clear. 

Please add at the conclusion your personal opinion about the gut microbiota role in the future treatment approach in acute ischemic stroke. 

Please add a specific personal opinion and comments about the possible correlation between atrial fibrillation and gut microbiota 

Author Response

  1. We have further elaborated on the role of neuroinflammation via the microbiota-gut-brain axis in the pathogenesis and prognosis of acute ischemic stroke (AIS), emphasizing the crucial role of neuroinflammation in reperfusion injury after recanalization therapy. (Page 2, line 56)
  2. We have provided a more in-depth discussion on the potential of gut microbiota as a novel treatment for AIS, and in the conclusion, we have suggested the clinical application of utilizing gut microbiota to improve the prognosis of AIS patients in the future. (Page 10, line 348, line 384)
  3. We have added a discussion on the association between atrial fibrillation and gut microbiota due to the subgroup of patients with severe disability has a higher proportion of atrial fibrillation. (Page 9, line 333)
  4. We have made the images consistent, and adjusted the color contrast for Figure 4 for better clarity and ensured that all figures meet the journal's quality standards.

Reviewer 2 Report

I highly recommend improving the quality of the Figures before publication.

It would be nice to justify in the paper why patients younger than 20 were explicitly excluded from the analsysis. Are there any differences in gut microbiota composition between yound and adult with stroke?

Author Response

  1. We have made the images consistent, and adjusted the color contrast for Figure 4 for better clarity and ensured that all figures meet the journal's quality standards.
  2. We have explained why patients under 20 years of age were excluded from the study. This was mainly due to ethical considerations. In Taiwan, minors under the age of 20 belong to the vulnerable group/special group in the review of research ethics and Institutional Review Board, and the presence of a legal representative is required to participate in clinical research. However, because of recanalization therapy is time-limited, it would be challenging to obtain consent from the legal representative of minors within the limited timeframe for recanalization therapy. In addition, AIS patients under the age of 20 are very rare. Therefore, patients under the age of 20 were excluded from this study.